# Programmable integrated photonics for topological Hamiltonians

Mehmet Berkay On[1,2], Farshid Ashtiani [1], David Sanchez-Jacome [3], Daniel Perez-Lopez [3], S. J. Ben Yoo[2] & Andrea Blanco-Redondo [1,4] ✉

A variety of topological Hamiltonians have been demonstrated in photonic platforms, leading to fundamental discoveries and enhanced robustness in applications such as lasing, sensing, and quantum technologies. To date, each topological photonic platform implements a specific type of Hamiltonian with inexistent or limited reconfigurability. Here, we propose and demonstrate different topological models by using the same reprogrammable integrated photonics platform, consisting of a hexagonal mesh of silicon Mach-Zehnder interferometers with phase shifters. We specifically demonstrate a one-dimensional Su-Schrieffer-Heeger Hamiltonian supporting a localized topological edge mode and a higher-order topological insulator based on a two-dimensional breathing Kagome Hamiltonian with three corner states. These results highlight a nearly universal platform for topological models that may fast-track research progress toward applications of topological photonics and other coupled systems.

The field of topological photonics[1,2] has gained tremendous traction in the last 15 years thanks to its unraveling of novel fundamental phenomena in topological physics as well as its potential to deliver robustness against certain types of defects and disorder for integrated photonic devices[3,4] such as lasers[5–8] and quantum information platforms[9–15]. The origins of topological photonics stem from the discovery of topological insulators in condensed matter physics[16,17], where materials that are insulating in their bulk can conduct electricity without dissipation on their edges. These concepts were translated into photonics platforms[18,19], where topology refers to a quantized property that describes the global behavior of the wavefunctions in a dispersion band. A key feature of topological photonics is the existence of modes that live on the edge of photonic materials with nontrivial topologies and that show resilience to certain types of disorder. These edge modes have been demonstrated in a variety of platforms, from one-dimensional (1D) arrays of waveguides[20–22] or resonators[23] with chiral symmetries, to two-dimensional (2D) lattices of helical waveguides[24] and ring resonators with asymmetric couplings[25], all the way through bianisotropic metamaterials[26] and quasicrystals[27]. While the majority of topological photonics platforms presented to date

have a static character, a number of reconfigurable topological photonic insulators have been experimentally realized in the last few years[28–30], as well as analogous concepts in acoustics[31,32] and plasmonics[33]. However, the reconfigurability in these platforms is limited to rerouting the pathways followed by the guided waves or switching these pathways on and off, while the type of Hamiltonian implemented in a given physical platform is fixed.

In parallel, programmable integrated photonic platforms have enabled fast development of a wide range of circuit architectures through real-time reconfiguration of a general-purpose photonic circuit via software programming[34]. Such systems typically consist of a 2D mesh of silicon photonics Mach–Zehnder interferometers (MZIs) whose transfer matrix can be programmed by adjusting the embedded phase shifters. This enables the reconfiguration of light paths through the mesh and the implementation of linear optical operations by interfering signals from different paths[35], showing a ground-breaking potential for communications, machine learning[36] and quantum information processing[37] among other applications.

Here, we propose and experimentally demonstrate that topological physics can be observed in programmable integrated photonics

[1]Nokia Bell Labs, 600 Mountain Ave, New Providence, NJ 07974, USA. [2]University of California Davis, Department of Electrical and Computer Engineering, One Shields Avenue, Davis, CA 95616, USA. [3]iPronics Programmable Photonics, Avenida Blasco Ibanez 25, 46010 Valencia, Spain. [4]CREOL, The College of Optics and Photonics, University of Central Florida, Orlando, FL 32816, USA. ✉e-mail: andrea.blancoredondo@ucf.edu

platforms. Importantly, virtually any topological model can be implemented in programmable integrated photonic platforms that allow for exquisite reconfigurable control of the hopping strength and hopping phase between elements, as well as of the real and imaginary part of the onsite energies. To illustrate this, we use a commercial programmable platform (*iPronics' Smartlight Processor*) to show robust localization of edge modes in a dimer chain of resonators resembling the Su-Schrieffer-Heeger (SSH) model[38] and of higher-order topological modes (corner modes) in a 2D breathing Kagome lattice[39] of resonators. Reprogrammable silicon photonic meshes represent a nearly universal test-bed for topological photonics, including non-Hermitian topological photonics[40–42], that could greatly accelerate fundamental discoveries as well as the development of applications.

## Results
### Integrated programmable mesh
A schematic view of the programmable silicon photonics chip used in our experiments is shown in Fig. 1a. It consists of a hexagonal mesh of programmable unit cells (PUCs), where each PUC is formed by a $2 \times 2$ MZI with a thermo-optic phase shifter in each arm, as depicted in Fig. 1c[43]. The two optical inputs enter a 50/50 multimode interference (MMI) coupler followed by two thermo-optic phase shifters to adjust the optical phase shift of each arm. Another 50/50 MMI coupler combines the two phase-adjusted signals and provides the PUC outputs. By controlling the phases imparted on each arm $\theta_1$ and $\theta_2$ one can realize any $2 \times 2$ complex unitary transfer matrix

$$T(\theta_1, \theta_2) = e^{i\phi} \begin{bmatrix} \cos(\Delta) & -\sin(\Delta) \\ \sin(\Delta) & \cos(\Delta) \end{bmatrix} \tag{1}$$

with

$$\phi = \frac{\theta_1 + \theta_2}{2} \tag{2}$$

representing a common phase to the two output signals and

$$\Delta = \frac{\theta_1 - \theta_2}{2} \tag{3}$$

determining the power splitting ratio between signals. Therefore, by programming the phase settings of the mesh PUCs, the optical signal can be routed into desired paths and arbitrary photonic circuit configurations can be realized.

To approach the realization of topological physics in programmable meshes we reconfigure the programmable cells of the mesh to create lattices of ring resonators with carefully engineered resonant frequencies and coupling rate between them. Note that, thanks to its interconnection profile, the hexagonal mesh allows for the programming of optical cavities and better resolution when compared to alternative lattice mesh designs[43,44], and it is, therefore, better suited to implement topological Hamiltonians. The smallest possible ring resonator in this hexagonal mesh consists of six PUCs, as schematically depicted by the blue circumferences in Fig. 1a. The PUCs shared between adjacent rings are programmed to determine the coupling rate (and if desirable the coupling phase) between the two rings. The power in each ring can be monitored by tapping a small amount of the power out of the ring to a monitoring photodiode, as depicted by the blue arrows exiting the lattice.

In particular, we have chosen to implement two different models to demonstrate the potential of programmable photonics to explore topological physics: a 1D SSH model and a 2D breathing Kagome lattice. Due to the size and shape of the currently available hardware mesh, a rectangular arrangement of 72 PUCs shown in Fig. 1a, only the 1D SSH model could be experimentally tested in the hardware. Nonetheless, we have implemented the 2D Kagome in a realistic simulator[45] of the mesh and we highlight that the size and shape of the lattice is well within the scalability scope of current technology.

### 1D topological photonics in the programmable mesh
We start by implementing the simplest topological model, the dimer chain, also referred to as the SSH model[38], which relies on an alternate pattern of weak and strong coupling between sites and was demonstrated in optical experiments in 2009 in an optically-induced superlattice[20]. Since then, many optical implementations of the SSH have been proposed: from femtosecond laser written waveguides in glass[21] to silicon photonics waveguides[22], all the way to microwave resonators[23] and others. All of these demonstrations have shown little to none reconfigurability. Here, we implement the SSH model in a programmable mesh by arranging the mesh into a bipartite lattice of seven-ring resonators, as schematically depicted in Fig. 1b. The experimental realization of this model on the silicon photonics programmable platform is marked by blue circumferences in Fig. 1a.

The Hamiltonian describing this system of seven rings is given by

$$H = \left[ k_w \sum_{n \in \{1,3,5\}} a_n^\dagger a_{n+1} + k_s \sum_{n \in \{2,4,6\}} a_n^\dagger a_{n+1} \right] + H.c. \tag{4}$$

where $k_w$ and $k_s$ are the strong and weak coupling strengths between sites and $a_n^\dagger$ and $a_n$ are the creation and annihilation operators on site $n$.

The calculated eigenvalues of this lattice, embodied here by the resonant frequencies of the supermodes, are shown in Fig. 2a for three different combinations of $k_w$ and $k_s$. From here on, we refer to this

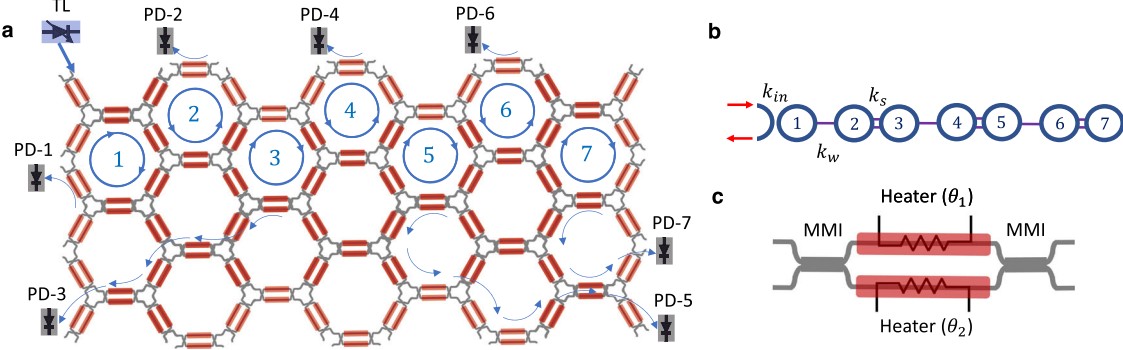

**Fig. 1 | Programmable integrated photonics platform and implementation of the SSH model. a** Schematic of the programmable mesh on iPronics' Smartlight Processor and reconfiguration for 7 coupled ring resonators; TL off-chip tunable laser, PD off-chip photodetector. **b** Implemented 1D SSH model. **c** Programmable unit cell in detail.

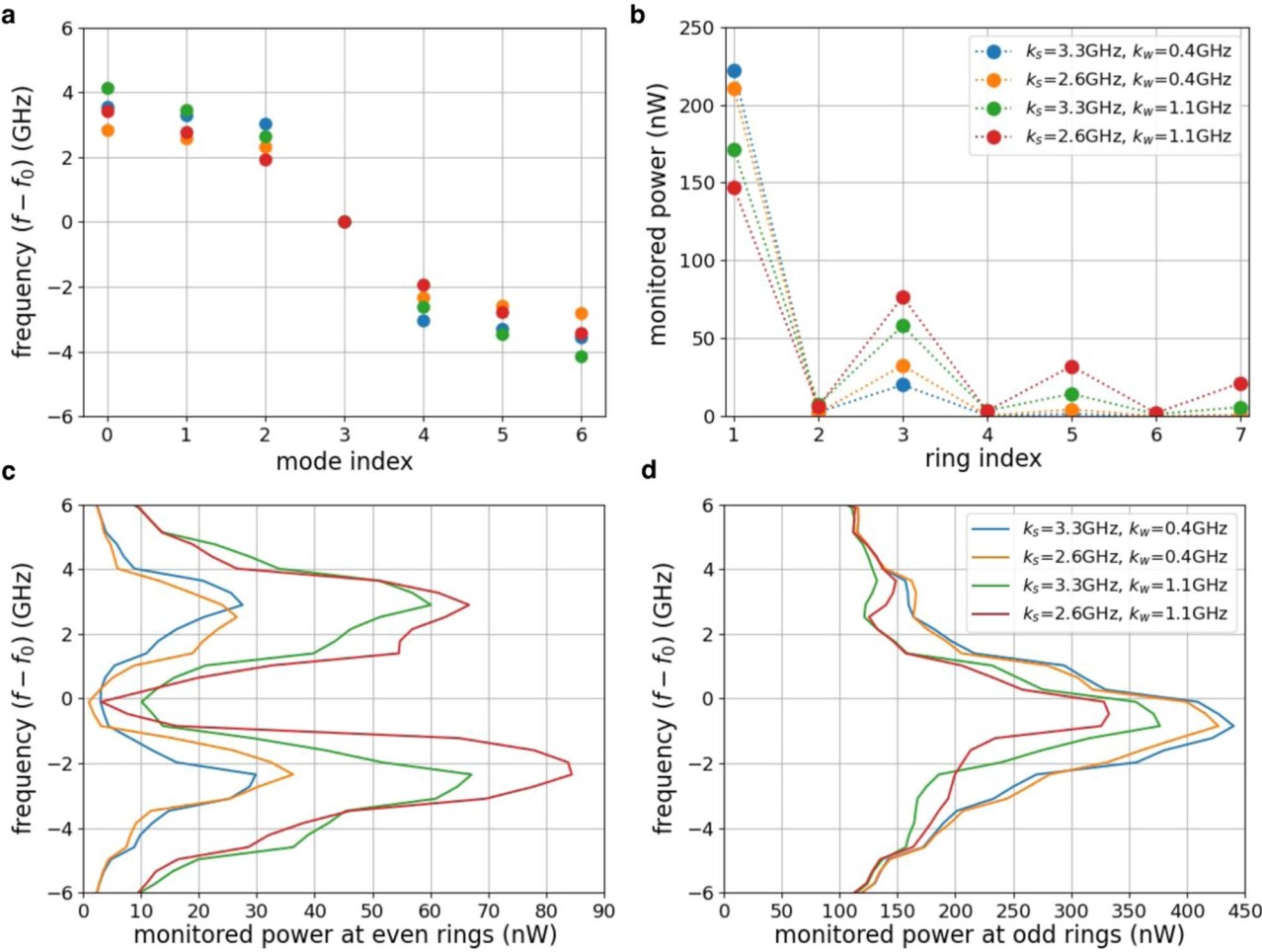

**Fig. 2 | Characterization of the eigenvalues and the topological edge mode in the SSH model. a** Calculated eigenvalues of the Hamiltonian matrices. **b** Monitored powers (tapped ≈ 1% of the power in each ring) at the coupled ring resonators from the programmable mesh hardware. **c** Total monitored power (tapped ≈ 1%) spectrum of the even rings of 1D SSH model. **d** Total monitored power spectrum of the odd rings of 1D SSH model.

alternate pattern of strong and weak coupling strengths as the dimerization pattern. The coupling strength can be accurately controlled by programming the PUC between every two rings, and it depends on the power splitting ratio of the common PUC and the frequency spectral range (FSR) between two rings (see Eq. 1 in Supplementary Note 1). The supermodes frequencies are offset to the resonant frequency of the individual ring resonators $f_0 = 193.396$ THz. Note that slight differences in $f_0$ between rings can be compensated by adjusting the phase shifters in the mesh (see Fig. S2 in Supplementary Note 1). This lattice is expected to have a band gap with a topological edge mode localized at ring 1. Stronger dimerization patterns, in other words stronger contrast between $k_w$ and $k_s$, are expected to lead to larger band gaps and consequently to stronger and more robust localization of the edge mode. Thus, the reprogrammability of the lattice lends us full control over the band gap, the degree of localization and the robustness of the edge mode.

To experimentally prove this, we connect a continuous wave tunable laser to the input port of the mesh and monitor the power in the rings under different conditions. First, we tune the laser wavelength to $f_0$ and monitor the power in each ring by tapping 1% of the power in each ring to a photodiode. The resulting measurements, shown in Fig. 2b, exhibit the characteristic modal distribution of the SSH edge modes with a maximum at the edge site and full localization in one of the sublattices, i.e., virtually zero power in the even rings. The measurements also confirm that stronger dimerization patterns lead to stronger localization at the edge, showing agreement with the

simulations, as shown in Fig. S3 in Supplementary Note 2. Subsequently, we tuned the input laser frequency within ± 6 GHz around $f_0$ and summed up the power in all the even and odd rings, as shown in Fig. 2c, d, respectively. By looking at the width of the dip around $f_0$ in Fig. 2c one can appreciate how the band gap grows with increasing dimerization strength. This is because the only supermode supported around $f_0$ is the topological edge mode, which is fully localized in the odd rings. The residual amount of power observed in the even rings at $f_0$ is due to the non-perfect overlap of the input light with the modal profile of the edge mode. Consequently, the peak exhibited around $f_0$ in the odd rings, as shown in Fig. 2d, correlates strongly with the power in the topological edge mode and it becomes higher with stronger dimerization. Further, the location of the peak in Fig. 2d represents the eigenvalue of the measured topological edge state and the two transmission bands in Fig. 2c correspond to the continuum of eigenvalues for the modes in the lower and upper bands. The measured eigenvalues match qualitatively well with the calculated eigenvalues for the programmed Hamiltonian Fig. 2a.

Next, we evaluate the robustness of the topological edge state by intentionally introducing perturbations on the coupling strengths. Specifically, twenty random variations are drawn independently from a normal Gaussian distribution around the nominal coupling strength for each pair of rings, $\sim \mathcal{N}(0, \sigma^2)$, where $\sigma$ is the standard deviation. Figure 3 shows the power in each ring at $f_0$ for two dimerization patterns – a *strong dimerization* case with $k_s = 3.3$ GHz, $k_w = 0.4$ GHz in Fig. 3a, b; and a *weak dimerization* case with $k_s = 3.3$ GHz and

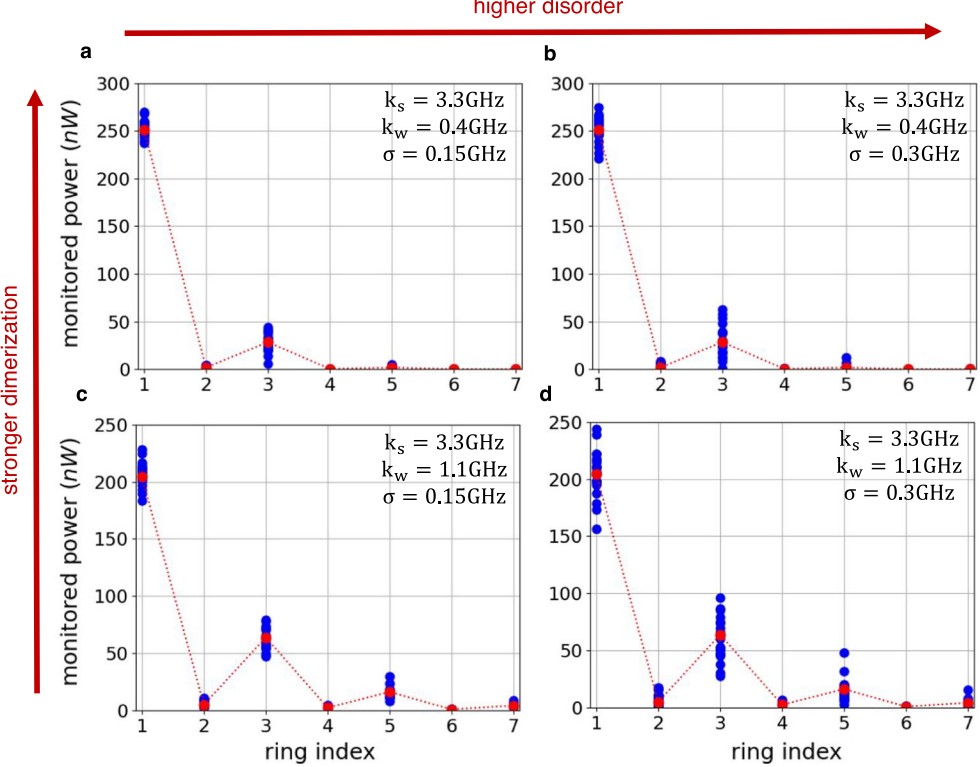

**Fig. 3 | Characterization of the SSH edge mode robustness to disorder.** Monitored powers (tapped ≈ 1%) at the coupled ring resonators from the programmable mesh hardware with 20 different random perturbation added on the coupling rates for **a** and **b** the *strong dimerization* case and **c** and **d** the *weak dimerization* case.

$k_w = 1.1$ GHz in Fig. 3c, d. For each case, we consider two levels of disorder – *low disorder* with $\sigma = 0.15$ GHz in Fig. 3a, c and *high disorder* with $\sigma = 0.3$ GHz in Fig. 3b, d. The red dots represent the power in each ring in the absence of deliberately introduced disorder and the blue dots represent the power in the rings when each of the twenty random iterations of disorder is implemented.

We can now quantify the robustness of the topological mode by measuring the standard deviation of the power in the rings under disorder in the coupling. For instance, under *low disorder* (*high disorder*) the standard deviation of the power in ring 1 is $\sigma_{power}^{ring-1} = 8.2$ nW (13.2 nW) in the *strong dimerization* case and 10.5 nW (20.4 nW) *weak dimerization* case. Since a strong signature of topological protection on the SSH model is the localization of light in one of the sublattices, we can also quantify the variation of the power in the even rings in the presence of disorder, which remains very close to zero in the *strong dimerization* case ($\sigma_{power}^{even-rings} = 1.3$ nW and 2.4 nW for low and high disorder respectively) and it becomes slightly larger in the *weak dimerization* case ($\sigma_{power}^{even-rings} = 2.1$ nW and 4.7 nW for low and high disorder respectively). Note that for exceedingly high levels of disorder of $\sigma \geq 1.5$ GHz the chiral symmetry breaks and that leads to the closing of the topological band gap and the disappearance of the edge mode.

As opposed to conventional topological photonic platforms in which a proper robustness study would require the fabrication and measurement of a large number of devices, this platform allows for accurate quantification of the robustness against disorder in the coupling on the same chip by just software reprogramming.

## 2D topological photonics in the programmable mesh

To illustrate the versatility of programmable integrated platforms in the context of topological photonics, we now implement a higher-order topological insulator (HOTI) based on a breathing kagome lattice. The kagome lattice is a 2D model consisting of corner sharing triangles with opposite orientations. While the tight-binding model of the kagome lattice exhibits graphene-like Dirac bands, a band gap opens when the coupling strengths between the sites in different triangles alternate. This is known as the breathing kagome lattice which has been shown to host higher-order topological corner states in a variety of settings[39,46–48], including photonics[49–53]. Here, we implement a fully reprogrammable breathing Kagome lattice by reconfiguring the silicon photonics mesh into a 2D array of coupled ring resonators arranged in corner sharing triangles with the upward pointing triangles and the downward pointing triangles having different coupling strengths, as depicted in Fig. 4b. The implementation of such 2D lattice requires 72 PUCs, exactly the number of PUCs available in the silicon photonics chip of our experiments, see Fig. 1a. However, the rectangular shape of this specific chip prevents the implementation of the model in Fig. 4b directly on the hardware, and thus we have implemented this model on a realistic simulator of the mesh[45]. Note that the scalability required for this demonstration is perfectly within the possibilities of the current technology.

The tight-binding Hamiltonian describing the breathing kagome lattice is

$$H = \left[ k_w \sum_{\langle n,m \rangle \in \triangle} a_n^\dagger a_m + k_s \sum_{\langle n,m \rangle \in \triangledown} a_n^\dagger a_m \right] + H.c. \qquad (5)$$

where $\triangle$ and $\triangledown$ represent the sites in the upward and downward pointing triangles. The theoretical energy spectra for three different dimerization patterns are shown in Fig. 5a. We observe three quasi-degenerate energies at $f - f_0 \approx 0$ that correspond to the energies of the corner states. The power distribution of one of those eigenmodes is shown in Fig. 5b, c for the stronger and weaker dimerization cases, respectively. It is evident that stronger dimerization leads to stronger light localization at the corners of the lattice.

Next, we simulate the insertion of light in ring 1 and monitor the power in each ring at the edge of the lattice. In the 2D Kagome lattice, light propagates clockwise and counter-clockwise directions inside the

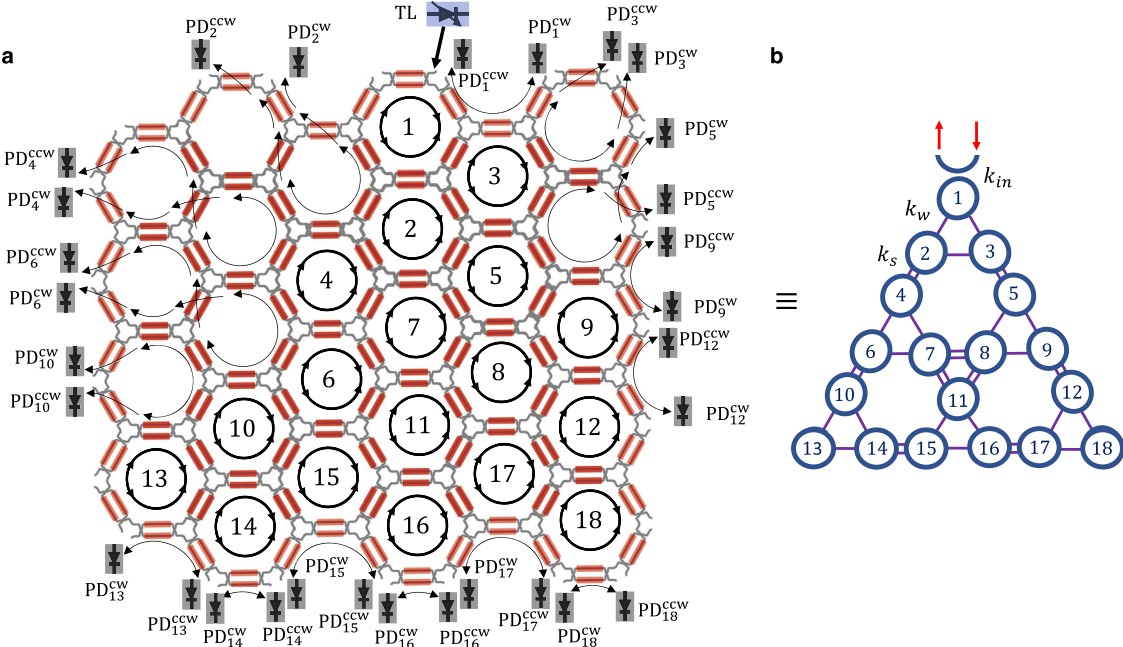

**Fig. 4 | Implementation of the breathing Kagome lattice in a programmable integrated photonics platform. a** Schematic of the programmable mesh on the simulator and reconfiguration for 18 coupled ring resonators, TL off-chip tunable laser, PD$^{(c)cw}$ clockwise and counter-clockwise off-chip photodetectors. **b** Implemented 2D breathing Kagome lattice.

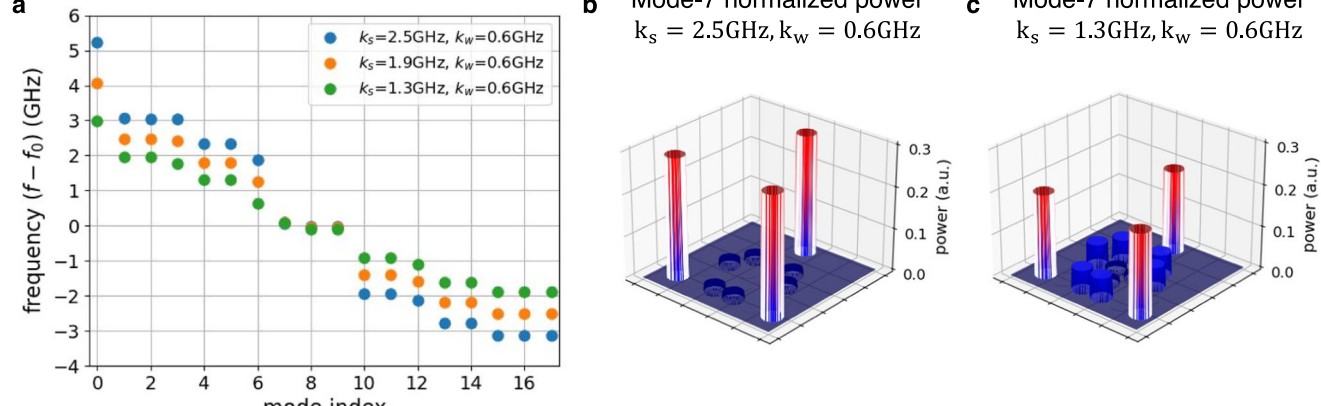

**Fig. 5 | Calculated eigenvalues and corner state in the breathing Kagome lattice. a** Energy spectra of the breathing Kagome lattice for three different dimerization patterns: (blue) $k_s = 2.5$ GHz and $k_w = 0.6$ GHz, (yellow) $k_s = 1.9$ GHz and $k_w = 0.6$ GHz, and (green) $k_s = 1.3$ GHz and $k_w = 0.6$ GHz; **b** Normalized power distribution of one of the quasi-degenerate corner states (mode-7) for $k_s = 2.5$ GHz, $k_w = 0.6$ GHz and **c** for $k_s = 1.3$ GHz, $k_w = 0.6$ GHz.

resonator, unlike the 1D SSH implemented on hardware mesh. Therefore, each resonator requires two monitoring ports and external detectors, as shown in Fig. 4a. First, we vary the input frequency of the laser within a range of $\pm 2$ GHz around $f_0$ and sum the monitored power in all the edge rings for each frequency, as shown in Fig. 6a. In the case with stronger dimerization (blue line) in Fig. 6a we observe a well-defined peak at $f_0$, which indicates that most of the input light populates the corner states and that these states are strongly degenerate. This is confirmed by the power distribution over the edge rings shown in 6b, in which the power is strongly localized in the three corner rings. Note that we do not have access to monitoring the bulk rings (rings 7, 8, and 11) for the current programmable mesh architecture. However, it is possible to implement monitoring inside the mesh by non-invasive, contactless integrated light probes[54].

Another interesting physical effect occurring in HOTIs under certain conditions is that of light *fractionalization* between the higher-order states[53]. Given the frequency degeneracy of the three corner states, inputting light in one of the corners is equivalent to exciting an equal superposition of the three corner eigenstates. We can observe some *fractionalization* of light to all three corners, in Fig. 6b, although the power in all three corners is not exactly equal. We have verified that this is due to the path-related phase differences experienced by the light reaching the bottom left and bottom right corners because of the slightly asymmetric implementation of the lattice in the silicon photonics mesh. This can be remediated by implementing a symmetric mesh, as demonstrated in Figs. S4 and S5 in Supplementary Note.

Subsequently, we focus our attention on the cases with moderate (yellow line) and weak (green line) dimerization in Fig. 6a. As the dimerization becomes weaker, so does the degeneracy of the corner states around $f - f_0 \approx 0$ and this translates into the two sub-peaks observed on the power spectrum monitored on the edge rings. This becomes more pronounced for the weakest dimerization case (green

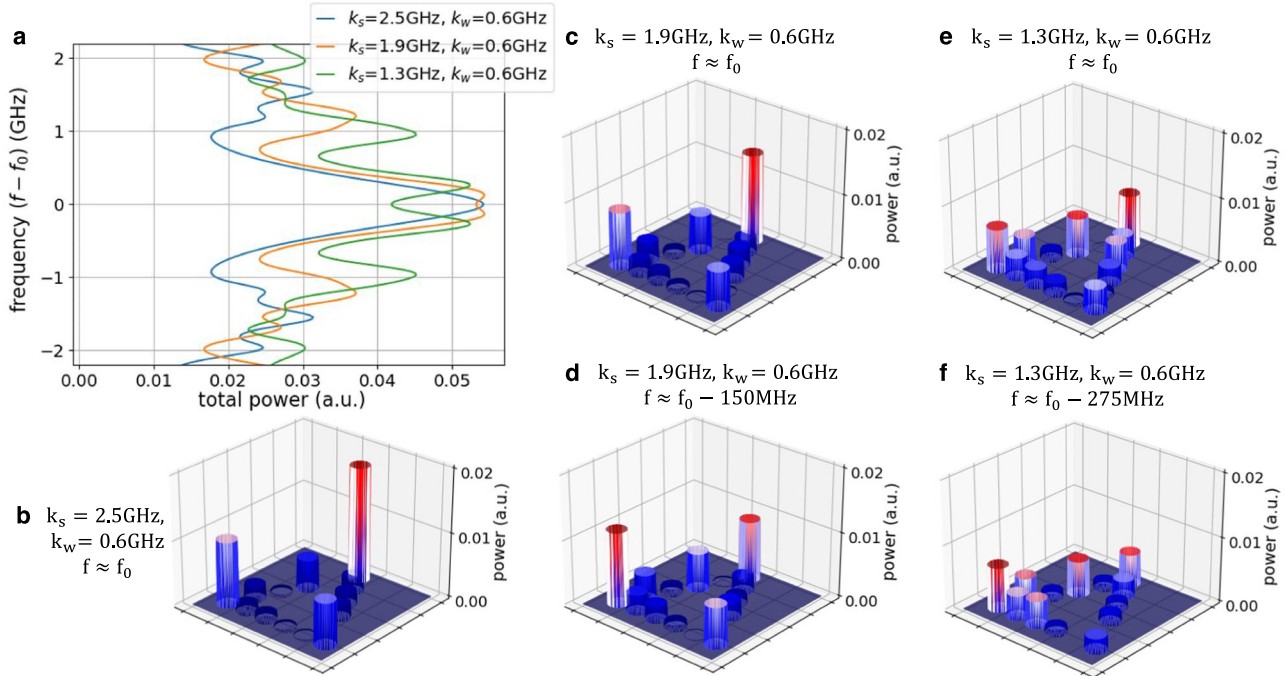

**Fig. 6 | Simulated excitation of the corner states in the breathing Kagome lattice. a** Simulated spectrum with various dimerizations, **b–f** Simulated power distribution on the Kagome lattice with various dimerizations and frequencies.

line). Therefore, as illustrated in Fig. 6c–f, when the input light has a frequency of $f_0$ the localization on the corner rings is not as strong as at the frequency of the subpeaks ($f_0 − 0.15$ GHz and $f_0 − 0.275$ GHz for the moderate and weak dimerization cases, respectively). For a quantifiable comparison, the percentage of light in the corner rings increases 36% when moving from $f_0$ to $f_0 − 0.275$ GHz in the weakest dimerization case.

## Discussion

We have proposed and demonstrated that programmable integrated photonics can be used to implement different topological photonics models and to fully reconfigure the behavior of topological modes. In the same platform we have implemented a 1D SSH chain and a 2D HOTI and we have shown full control over the localization and robustness of the edge and corner modes.

The possibility of engineering, not only the coupling rate between sites, but also the phase of such couplings renders this platform readily available for the implementation of a wide variety of topological models, including magnetic-like Hamiltonians that have shown potential in lasers[7] and quantum optics functionality[9,12]. Moreover, the loss of each ring can also be individually and accurately controlled, opening a plethora of possibilities for non-Hermitian topological photonics investigations and devices[40–42,55].

Another enticing future research avenue on this kind of programmable integrated platform is the exploration of lattices with explicitly broken time reversal symmetry ($T$) by time-harmonic modulation of the coupling strength between resonators[56]. A crucial requirement here is that the strength of the modulation must be larger than the decay rate, which translates into the need for fast modulation and low loss technologies. While the current hardware uses heaters to control the coupling and the loss is relatively high, it is within the scalability scope of this technology to introduce high-speed electro-optics phase shifters and significantly reduce the loss of each cell. This would open the door to the study of a variety of truly non-reciprocal systems at optical frequencies with important fundamental and practical implications.

By showing that a general-purpose programmable integrated photonics platform can be used to implement nearly any topological

photonics model we hope to accelerate progress in the field, bypassing lengthy design and fabrication cycles and offering a fully reconfigurable platform in which the topological modes are easily tailored and the effects of disorder can be accurately quantified.

## Methods
### Processor and experimental setup
The iPronics SmartLight processor is a silicon photonics programmable platform that consists of a hexagonal mesh of 72 programmable unit cells (PUCs). Each PUC is formed by an 811 μm long 2 × 2 MZI with an average insertion loss of 0.5 dB. The output of a tunable laser is coupled to the mesh with about 2–4 dB fiber to chip coupling loss and an additional 6–8 dB on-chip routing loss from the input coupler to the desired PUC. The laser is thermally tunable from 1549.9 nm to 1550.2 nm with a resolution of 3 pm. Optical measurements are performed using off-chip PDs with a sensitivity of −70 dBm. The photonic chip is placed on a thermo-electric cooler (TEC) for temperature control. For a room temperature variation of ± 6 °C, the chip temperature is stable within ± 0.5 °C of its set-point which can be compensated for during the system phase calibration. The status of each PUC (that is the coupling rate) can be individually programmed through a Python interface to realize the desired topological configuration in real time.

### Simulation platform
The simulation platform can be used to implement larger and arbitrary shape configurations such as a 2D Kagome lattice. The simulator consists of 198 PUCs that can be individually programmed similar to the hardware mesh. The loss and initial phase of all PUCs can be set for a realistic simulation. In the case of the 2D Kagome lattice, each PUC has a loss of 0.1 dB which is technologically feasible. The input laser source is tuned with a resolution of 0.2 pm around 1550 nm.

## Data availability
All experimental and simulation data supporting the findings are presented in the paper and Supplementary Information in graphic form.

Source data will be provided by the corresponding authors upon request.

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

## Acknowledgements

A.B.-R. is supported by the National Science Foundation (NSF) (award ID 2328993).

## Author contributions

A.B.-R., M.B.O, and F.A. conceived the experiment and simulations, analyzed the results, and wrote the paper. M.B.O. performed the experiment and simulations. D.S.-J. and D.P.-L. built the experimental setup, developed the required software for the experiment, provided technical support, and edited the paper. S.J.B.Y. contributed to technical discussions and edited the paper.

## Competing interests

The authors declare no competing interests.
