## [Peer Review File · Nature Communications]

REVIEWER COMMENTS

Reviewer #2 (Remarks to the Author):

The draft titled “Programmable Integrated Photonics for Topological Hamiltonians” proposed and experimentally demonstrated the approach to realize topological Hamiltonians by using the commercially available programmable integrated photonics circuits, which is a big leap for the topological photonics research. In the draft, the authors used iPronics’ Smart-light Processor showing the 1D topological model, Su-Schrieffer-Heeger (SSH) model experimentally. However, they did not show the higher order topological insulator based on a breathing Kagome lattice due to the limited PUC number in the iPronics’ Smart-light Processor. Instead, they used numerical simulator to demonstrate it. Based on the careful reading of the draft, I would recommend it to be published in Nature Communications if they can address the following comments.

1. In the main text and supplementary, authors mentioned about the calibration of the processor. Is there need to calibrate the processor during the measurement procedure since there will be temperature fluctuation with different micro heaters turned on in the chip? How strong of the temperature effect would be on the demonstration of topological Hamiltonians?
2. The insertion loss of the processor looks high, $\sim 30\text{dB}$. Will the high insertion loss affect the generation of the topological edge mode or corner states?
3. There is no clear instruction on how to define the different dimerization patterns. It would be great if the author could supply the definition.
4. In figure 5 and 6, the power distribution difference for different dimerization patterns is not clear. It would be great if the illustration can be clearer either by using the same scale bar or using different size of the circles. Besides, it would be great if the author could put scale bar legend on the figures.

Reviewer #3 (Remarks to the Author):

The manuscript 'Programmable Integrated Photonics for Topological Hamiltonians' shows realization of topological models with programmable photonic processors.

In particular it is shown how the dimerized SSH chain and breathing Kagome lattice can be realized in photonic hardware.

The authors demonstrate how lengthy fabrication and design cycles can be skipped by using a programmable platform, in particular for the case of parameter sweeps and average over disorder configuration. The field of applications is broad ranging from lasers over quantum optics to investigation of non-Hermitian physics and time reversal broken systems.

In my view, the manuscript could be significantly improved by providing clarifications to above points:

-Is it possible to compare exact diagonalization (ED) on the programmed tight binding Hamiltonian and experiment more to show the degree of control one has with the photonics hardware?

In particular, can one compare the penetration depth of the topological edge/corner states from the numerics to the one calculated from the experimental data?

How are the discrete eigenmodes appearing in Fig. 2a) related to the data shown in Fig. 2c) + d) (local density of states (LDOS) on even/ odd sites)?

-The chiral symmetry is not exact, Fig. 2(d) shows this (small amount power on even sites) and there is a discussion in the text.

There is a length scale associated with chiral symmetry breaking that limits the device length (after which one crosses over into a topologically trivial state, with no even/odd distinction), it would be useful to provide an estimate.

Further mild criticism and comments:

The system sizes were relatively few sites (<100), relatively high loss due to the use of heaters (does this limit the system size here or could this hardware be bigger in principle?) and chiral symmetry responsible for topological protection against disorder is not exact.

The Kagome lattice could not be implemented on the available hardware and the data is obtained by simulations.

The methods used are state of the art (novel programmable photonic platform / simulation thereof).

The data were obtained by using a commercially available platform (iPronics) and parameters given in the manuscript and supplemental material allow to reproduce it. Details on calibration procedures and additional consistency checks are provided. The sketches of the programmed lattices with measurement overhead are instructive.

REVIEWER COMMENTS

Reviewer #2 (Remarks to the Author):

The draft titled “Programmable Integrated Photonics for Topological Hamiltonians” proposed and experimentally demonstrated the approach to realize topological Hamiltonians by using the commercially available programmable integrated photonics circuits, which is a big leap for the topological photonics research.

We thank the reviewer for the time devoted to this careful review and for the thoughtful comments and suggestions that we address one by one below.

In the draft, the authors used iPronics’ Smart-light Processor showing the 1D topological model, Su-Schrieffer-Heeger (SSH) model experimentally. However, they did not show the higher order topological insulator based on a breathing Kagome lattice due to the limited PUC number in the iPronics’ Smart-light Processor. Instead, they used numerical simulator to demonstrate it. Based on the careful reading of the draft, I would recommend it to be published in Nature Communications if they can address the following comments.

1. In the main text and supplementary, authors mentioned about the calibration of the processor. Is there need to calibrate the processor during the measurement procedure since there will be temperature fluctuation with different micro heaters turned on in the chip?

The processor does not need to be calibrated during the measurements, despite the temperature fluctuations mentioned by the reviewer. Let us elaborate on this answer.

First, we must clarify that there are two kinds of calibration involved in this system:

There is an initial calibration that it is performed once, as part of the processor first validation process. The calibration extracts the passive offset of each programmable unit cell that arises due to design and fabrication errors. This data is stored in the logic unit of the processor and remains fixed in time and for a considerable $\pm 5^{\circ}\text{C}$ chip base temperature range therefore there’s no need for further calibration of the device after the first routine. This calibration procedure obtains the phase-to-current mapping for each phase actuator enabling a very fine software-defined tuning of each actuator’s resultant phase. This data is reliable for long term process and also fixed for a considerable chip-based temperature range.

The calibration described in the paper pertains to the specific configuration of the mesh in arrays of coupled ring resonators. Although ideally all programmed ring resonators consists of six programmable unit cells (PUCs) and should have the same resonant frequency, small fabrication imperfections and temperature differences can lead to misalignments between the resonances of each ring. Specifically, despite the temperature of the processor being controlled via a thermo-electric cooler (TEC), the chip can undergo temperature drifts of $\pm 0.5^{\circ}\text{C}$ due to large temperature room fluctuations of $\pm 6^{\circ}\text{C}$ (e.g. air conditioning on and off). Such temperature drifts could translate in ± 5 pm shifts in the resonant frequency of a ring with 6

PUCs. Therefore, before we take our measurements, we adjust the phase ϕ of each PUC to correct for this misalignment in the resonant frequencies of the programmed rings. This does not need to be repeated during the measurements.

Finally, as pointed by the reviewer, thermal crosstalk from the neighboring cells could produce a residual temperature increase that could lead to undesired heat and phase deviations. These have been measured, modeled and analyzed in [1], confirming that the under-etched architecture of each phase actuator ensures marginal contribution of the thermal crosstalk when circuits with low driving-phase requirements are synthesized on the mesh. For more complex cases, the model in [1] can be employed to counter-act the effect without recalibration. In the present work no recalibration is done during the measurements.

[1] Cem, A., Sanchez-Jacome, D., Pérez-López, D., and Da Ros, F., "Thermal crosstalk modeling and compensation for programmable photonic processors," in [IPC], accepted (2023). Cem, A., Sanchez-Jacome, D., Pérez-López, D., and Da Ros, F., "Thermal crosstalk modeling and compensation for programmable photonic processors," in [IPC], accepted (2023).

We now clarify this in the text of the supplementary note 1:

"All the hexagonal resonators programmed in the mesh consist of six ideally equal PUCs and should, therefore, have equal resonant frequencies. However, fabrication variations on the silicon waveguide cause phase errors [1]. Additionally, local temperature variations on the processor chip shift the resonance wavelength of the individual resonators [2]. The processor is calibrated once as part of its first validation process. The calibration extracts the passive offset of each programmable unit cell that is generated due to design and fabrication errors. This data is stored in the logic unit of the processor and remains fixed in time and for a considerable $\pm 5^\circ\text{C}$ chip base temperature range therefore there's no need for further calibration of the device after the first routine. Even though the first validation process compensates phase offset for each PUC individually, we observed that resonance wavelengths of the resonators vary, as shown in Fig.S2a because small fabrication imperfections and temperature differences lead to misalignment between the resonances of each ring which consists of six PUCs. Specifically, despite the temperature of the processor being controlled via a thermo-electric cooler (TEC), the chip can undergo temperature drifts of $\pm 0.5^\circ\text{C}$ due to large temperature room fluctuations of $\pm 6^\circ\text{C}$ (e.g. air conditioning on and off). Such temperature drifts could translate in $\pm 0.5\text{pm}$ shifts in the resonant frequency of a ring. By using two thermooptical phase shifters in the 2×2 programmable unit cell (PUC), (equation (1-3) in the main text), we can tune the resonance wavelength of the individual resonators without disturbing power coupling ratios. We run the calibration procedure once before the 1D SSH model measurements and do not recalibrate during the measurements."

How strong of the temperature effect would be on the demonstration of topological Hamiltonians?

If the individual resonant frequencies of the resonators were not aligned to the same value, there could be disorder of up ± 5 pm in the resonant frequencies of the individual resonators for large room temperature fluctuations. This can be viewed as disorder in the on-site energies of the microresonators, also referred to as on-diagonal disorder. How this affects the topological

Hamiltonian depends on the specific Hamiltonian. For instance, the SSH Hamiltonian is only strictly protected against off-diagonal disorder (disorder in the couplings) and therefore the effect of uncontrolled temperature fluctuations leading to on-diagonal disorder would be notable. That is why the calibration process is important in our experiments. In contrast, quantum spin Hall Hamiltonians are robust to on-diagonal disorder and the effect would be very small.

2. The insertion loss of the processor looks high, ~30dB. Will the high insertion loss affect the generation of the topological edge mode or corner states?

We would like to clarify the point that supplementary note Fig.S2, calibration figures show around -30dBm. However, this value does not directly correspond the total insertion loss of the system. The calibration is performed by looking monitored power after 1% power tapping by monitoring PUCs. Still, the reviewer's point is valid about the insertion loss and its effect on topological modes in the current platform. The insertion loss of the employed device is dominated by the chip-to fiber loss (2-4 dB), on-chip routing loss (6-8 dB) and the programmable unit cell loss (0.5 dB per cell). However, this does not affect the generation of the edge mode or the corner states, just the scalability to larger systems.

In future work we are targeting the progressive loss minimization to reach a total loss of 3 dB (full access) and 0.1 dB / PUC, this will allow the direct hardware implementation of larger two-dimensional topological lattices.

We now give specific insertion loss data in the main manuscript, Methods:

“Each PUC is formed by an 811 μ m long 2x2 MZI with an average insertion loss of 0.5dB. The output of a tunable laser is coupled to the mesh with about 2-4dB fiber to chip coupling loss and an additional 6-8dB on-chip routing loss from the input coupler to the desired PUC.”

3. There is no clear instruction on how to define the different dimerization patterns. It would be great if the author could supply the definition.

Thanks for pointing out that this may need further clarification. We now clarify this in the text (the text in red has been added in the revision):

“The calculated eigenvalues of this lattice, embodied here by the resonant frequencies of the supermodes, are shown in Fig. 2a for three different combinations of k_w and k_s . From here on, we refer to this alternate pattern of strong and weak coupling strengths as the dimerization pattern. The coupling strengths can be accurately controlled by programming the PUC between every two rings, and it depends on the power splitting ratio of the common PUC and the frequency spectral range (FSR) between two rings (see Supplemental Document Section 1 for the exact expression).”

4. In figure 5 and 6, the power distribution difference for different dimerization patterns is not

clear. It would be great if the illustration can be clearer either by using the same scale bar or using different size of the circles. Besides, it would be great if the author could put scale bar legend on the figures.

We appreciate this suggestion. To address this issue, we now use the same scale for all panels, and we also show the power distribution in 3D plots that show the differences between different dimerization patterns much more clearly. These are the new figures:

Fig 5: **a** Energy spectra of the breathing Kagome lattice for three different dimerization patterns: (blue) $k_s = 2.5$ GHz and $k_w = 0.6$ GHz, (yellow) $k_s = 1.9$ GHz and $k_w = 0.6$ GHz, and (green) $k_s = 1.3$ GHz and $k_w = 0.6$ GHz; **b** Normalized power distribution of one of the quasi-degenerate cornerstates (mode-7) for $k_s = 2.5$ GHz, $k_w = 0.6$ GHz and **c** for $k_s = 1.3$ GHz, $k_w = 0.6$ GHz

Fig 6: **a** Simulated spectrum with various dimerizations, **b-f** Simulated power distribution on the Kagome lattice with various dimerizations and frequencies.

Reviewer #3 (Remarks to the Author):

The manuscript 'programmable Integrated Photonics for Topological Hamiltonians' shows realization of topological models with programmable photonic processors. In particular it is shown how the dimerized SSH chain and breathing Kagome lattice can be realized in photonic hardware.

The authors demonstrate how lengthy fabrication and design cycles can be skipped by using a programmable platform, in particular for the case of parameter sweeps and average over disorder configuration. The field of applications is broad ranging from lasers over quantum optics to investigation of non-Hermitian physics and time reversal broken systems.

We thank the reviewer for the careful review of our manuscript and the constructive suggestions that we address point by point below.

In my view, the manuscript could be significantly improved by providing clarifications to above points:

-Is it possible to compare exact diagonalization (ED) on the programmed tight binding Hamiltonian and experiment more to show the degree of control one has with the photonics hardware?

We appreciate this useful suggestion.

To compare exact diagonalization on the ideal tight-binding Hamiltonian – that is the Hamiltonian that is built based on the ideal values for the resonant frequency of each ring and the coupling strengths – with the experimentally achieved Hamiltonian – the one built from the measured resonant frequencies and coupling strengths – one would need to have access to measure the coupling strengths, which we cannot do directly.

However, we can provide comparisons between the programmed eigenvalues and the measured data. Specifically, the peak of the measured power curves in Fig. 2d represents the measured eigenvalue of the topological mode, which can be compared with the eigenvalue of the topological edge mode for the programmed Hamiltonian in Fig. 2a. Further, the transmission bands in Fig. 2c, represent the continuum of eigenvalues for the measured extended modes in the lattice and can be compared with the lower and upper bands of the programmed eigenvalues in Fig. 2a.

We now explicitly draw this comparison in the text:

“Consequently, the peak exhibited around f_0 in the odd rings, as shown in Fig. 2d, correlates strongly with the power in the topological edge mode and it becomes higher with stronger dimerization. Further, the location of the peak in Fig. 2d represents the eigenvalue of the measured topological edge state and the two transmission bands in Fig. 2c correspond to the continuum of eigenvalues for the modes in the lower and upper bands. The measured eigenvalues match qualitatively well with the calculated eigenvalues for the programmed Hamiltonian Fig. 2a.”

In particular, can one compare the penetration depth of the topological edge/corner states from the numerics to the one calculated from the experimental data?

This is another great suggestion. Now, in the supplementary material we show a direct comparison between the penetration depth of the edge state of the simulated topological edge state and the measured topological edge state.

The new text and figure now read in supplementary note 2:

“We simulated the coupled ring resonators and compared simulation results with measurements of the 1D SSH model from the programmable hardware. The simulation parameters, insertion loss per PUC, laser wavelength, coupling, and monitoring PUC locations are set accordingly to match the hardware implementation. Fig.S3a, b presents power distribution at the resonance wavelength from the simulations and measurements. We fitted the function $f(x) = c_1e^{-\alpha x} + c_2$ on the power values of the odd rings. Here, x is the odd rings’ indices. The parameter α refers to the penetration depth of the edge states. “

Fig S3: Powers at coupled ring resonators from **a** simulator and **b** measured on hardware. Dashed traces are fitting function $f(x) = c_1e^{-\alpha x} + c_2$ on the odd rings of 1D SSH model.

How are the discrete eigenmodes appearing in Fig. 2a) related to the data shown in Fig. 2c) + d) (local density of states (LDOS) on even/ odd sites)?

The discrete eigenmodes in Fig. 2a are calculated by diagonalization of the Hamiltonian. Fig. 2c and Fig. 2d show the added measured power on all even rings and all the odd rings, respectively, when the input frequency is swept from $f_0 - 6\text{GHz}$ to $f_0 + 6\text{GHz}$. There is a strong correlation between the measurements in Fig. 2c and Fig. 2d and the eigenmodes in Fig. 2(a) as described above. Regarding the apparent discrete character of the eigenmodes in Fig. 2(a) vs the continuous nature of Figs. 2c and Fig. 2d this is related to the fact that the eigenvalues of the modes in the bands are actually very close to each other. This combined with these modes having a finite bandwidth means that the transition between two neighboring eigenstates when the input frequency is swept is smooth and one cannot distinguish between two contiguous modes in the band.

-The chiral symmetry is not exact, Fig. 2(d) shows this (small amount power on even sites) and there is a discussion in the text.

The reviewer is correct at pointing out that there is a small amount of power on the even rings observable in Fig. 2c. This power is very small - at the frequency of the edge mode, f_0 , the power in the even rings is $\sim 0.5\%$ of the power in the odd rings.

This residual amount of power at the even rings at f_0 is not related to the chiral symmetry not being exact but to the fact that the excitation of the edge mode is not perfect. The exclusive excitation of the topological edge state would require an input amplitude profile that overlaps perfectly with the topological eigenvector. Since this is not possible, the structure is excited only in ring 1, that has strong support for the edge mode; however, the overlap between the input light and the modal distribution of the edge mode is not 100%, which leads to a small amount of input power not coupled to the edge mode and spreading through the lattice.

We now clarify this in the text:

“This is because the only supermode present supported around f_0 is the topological edge mode, which is fully localized in the odd rings. The residual amount of power observed in the even rings at f_0 is due to the non-perfect overlap of the input light with the modal profile of the edge mode.”

There is a length scale associated with chiral symmetry breaking that limits the device length (after which one crosses over into a topologically trivial state, with no even/odd distinction), it would be useful to provide an estimate.

In the absence of disorder there is no chiral symmetry breaking. The introduction of increasing levels of disorder will eventually lead to chiral symmetry breaking and in turn to the closing of the topological band gap and to the transition to trivial states, as the reviewer points out.

We now add this information in the text:

“Since a strong signature of topological protection on the SSH model is the localization of light in one of the sublattices, we can also quantify the variation of the power in the even rings in the presence of disorder, which remains very close to zero in the strong dimerization case ($\sigma_{\text{even-rings power}} = 1.3\text{nW}$ and 2.4nW for low and high disorder respectively) and it becomes slightly larger in the weak dimerization case ($\sigma_{\text{even-rings power}} = 2.1\text{nW}$ and 4.7nW for low and high disorder respectively). Note that for exceedingly high levels of disorder of $\sigma \geq 1.5\text{GHz}$ the chiral symmetry breaks and that leads to the closing of the topological band gap and the disappearance of the edge mode.”

Further mild criticism and comments:

The system sizes were relatively few sites (< 100), relatively high loss due to the use of heaters (does this limit the system size here or could this hardware be bigger in principle?) and chiral symmetry responsible for topological protection against disorder is not exact.

The mesh could be scaled up. The key scaling limit is optical loss, as we are not area-limited nor packaging-limited. In this sense, the insertion loss of the employed device is dominated by the chip-to fiber loss (2-4 dB), on-chip routing loss (6-8 dB) and the programmable unit cell loss (0.49 dB per cell). In a future work we are targeting the progressive loss minimization to reach a total loss of 3 dB (full access) and 0.1 dB / PUC., allowing the proof of work of more complex topological photonics cases. A mesh as the one simulated for the Kagome lattice could be fabricated. Larger meshes integrating 2000 PUCs will be part of future works, representing a major technologic step forward.

The Kagome lattice could not be implemented on the available hardware and the data is obtained by simulations.

This is correct. But we would like to point out that this is not a fundamental limitation of the current technology. The Kagome lattice required 72 programmable unit cells, which is the same size as the current lattice. The issue was that the shape of the specific hardware platform in our lab is rectangular (with more cells in the horizontal than in the vertical direction). If the platform had a squarer configuration, we could have implemented the Kagome lattice experimentally.

The methods used are state of the art (novel programmable photonic platform / simulation thereof).

The data were obtained by using a commercially available platform (iPronics) and parameters given in the manuscript and supplemental material allow to reproduce it. Details on calibration procedures and additional consistency checks are provided. The sketches of the programmed lattices with measurement overhead are instructive.

REVIEWERS' COMMENTS

Reviewer #2 (Remarks to the Author):

The authors have answered my concerns very well.

I would recommend to accept it for publication.

Reviewer #3 (Remarks to the Author):

The authors response to the reviewers questions is adequate.

In particular, they clarify details on their calibration procedure, in how far temperature fluctuations affect topological protection and they provide an estimate of the insertion loss.

Figures for the 2D Kagome corner states were improved to emphasize the distinction of different dimerization patterns.

Further they present convincing additional data on the comparison of programmed and experimentally observed Hamiltonian. There is qualitative agreement of targeted and observed decay lengths into the bulk. The degree to which the chiral symmetry responsible for topological protection is approximate is discussed more clearly.

Therefore I recommend publication of the revised manuscript.